# Exploring the Role of a Putative Secondary Metabolite Biosynthesis Pathway in *Mycobacterium abscessus* Pathogenesis Using a *Xenopus laevis* Tadpole Model

**DOI:** 10.3390/microorganisms12061120

**Published:** 2024-05-31

**Authors:** Nicholas James Miller, Dionysia Dimitrakopoulou, Laurel A. Baglia, Martin S. Pavelka, Jacques Robert

**Affiliations:** Department of Microbiology and Immunology, University of Rochester Medical Center, Rochester, NY 14642, USA; nicholas_miller@urmc.rochester.edu (N.J.M.); d.dimitrak@hotmail.com (D.D.); laurel_baglia@urmc.rochester.edu (L.A.B.); martin_pavelka@urmc.rochester.edu (M.S.P.J.)

**Keywords:** *Xenopus*, macrophage recruitment, comparative immunology, intravital microscopy

## Abstract

*Mycobacterium abscessus (Mab)* is an emerging human pathogen that has a high rate of incidence in immunocompromised individuals. We have found a putative secondary metabolite pathway within *Mab*, which may be a key factor in its pathogenesis. This novel pathway is encoded in a gene cluster spanning MAB_0284c to 0305 and is related to *Streptomyces* pathways, producing the secondary metabolites streptonigrin and nybomycin. We constructed an in-frame deletion of the MAB_0295 (*phzC*) gene and tested it in our *Xenopus laevis* animal model. We have previously shown that *X. laevis* tadpoles, which have functional lungs and T cells, can serve as a reliable comparative model for persistent *Mab* infection and pathogenesis. Here, we report that tadpoles intraperitoneally infected with the ∆*phzC* mutant exhibit early decreased bacterial loads and significantly increased survival compared with those infected with WT *Mab*. ∆*phzC* mutant *Mab* also induced lower transcript levels of several pro-inflammatory cytokines (*IL-1β*, *TNF-α*, *iNOS*, *IFN-γ*) than those of WT *Mab* in the liver and lungs. In addition, there was impaired macrophage recruitment and decreased macrophage infection in tadpoles infected with the ∆*phzC* mutant, by tail wound inoculation, compared to those infected with the WT bacteria, as assayed by intravital confocal microscopy. These data underline the relevance and usefulness of *X. laevis* tadpoles as a novel comparative animal model to identify genetic determinants of *Mab* immunopathogenesis, suggesting a role for this novel and uncharacterized pathway in *Mab* pathogenesis and macrophage recruitment.

## 1. Introduction

*Mycobacterium abscessus* (*Mab)* is an opportunistic pathogen of the non-tuberculosis mycobacterium (NTM) group that tends to primarily infect human patients with immunocompromised conditions [1]. Chronic pulmonary infections with *Mab* tend to occur most commonly within individuals with major structural lung diseases, as well, such as cystic fibrosis or bronchiectasis [2]. The prevalence of NTM infections has risen in recent years, and the United States alone has seen over 175,000 annual cases, with a total burden of USD 1.7 billion dollars in medical funds [3]. Projections have indicated that there has been a continued increase in NTM lung diseases, specifically in the elderly and among women [4].

*Mab* infection brings significant clinical challenges regarding treatment. Several multidrug-resistant strains of *Mab* have become the primary clusters observed clinically [5,6,7]. Treating *Mab* infection tends to be extremely costly and requires a long cycle of a combination of antibiotics to be effective [2]. Another challenge of clinical *Mab* treatment is that *Mab,* throughout its infection, will spontaneously induce its rough (R) morphotype, which has a greater propensity to form biofilms, increasing its resistance to antibiotics [8]. The R morphotype is distinguished from the smooth (S) morphotype by a lack, or lower production, of glycopeptidolipid, which tends to promote increased bacterial survival within the host [9]. S Mab is typically found in the environment and during initial infection, while R Mab is more prominent in patients with severe pulmonary infections and appears to be more persistent [10,11]. However, the respective pathogenicity of the R and S morphotypes remains poorly understood, stimulating our investigation of other potential virulence factors.

In terms of host immune response, the components required for an appropriate immune response against NTM include a variety of cells and signals. These include TLR-2, TNF-alpha, T cells, neutrophils, and IFN-gamma. A unique aspect of IFN-gamma is that its excessive induction in certain *Mab* infections may allow for its pathogenesis [12]. *Mab* can reduce neutrophil activation in an attempt to promote its survival [13]. Macrophages are also a key component in the host defense against *Mab*. The challenge, however, is that many NTMs can circumvent the macrophage’s defense mechanisms and use them as a host, allowing for greater distribution throughout the tissues [3,14,15]. 

Previously, the most common animal models used to study *Mab* pathogenesis were mice and zebrafish [9]. However, both animal models possess limitations regarding their effectiveness. The murine model clears *Mab* infection rapidly, and the zebrafish model lacks functional lungs and fully functional adaptive immunity [3,9,16]. This challenge led our laboratory to develop a novel *Mab* infection model using the amphibian *Xenopus laevis*, which allows for bacterial infection to persist for up to 50 days, much longer than the time period of the other two models [17]. Other benefits of the *X. laevis* model are that they exhibit functional T cells and lungs during the tadpole and adult stages, which allows for better replication of both the infection model and the mode of defense in humans [17,18,19]. 

The analysis of the *Mycobacterium abscessus* ATCC 19977 (*Mab*) genome revealed the expected core mycobacterial genes, as well as genes that may have been obtained via horizontal gene transfer from other soil organisms, notably other actinobacteria [20]. Some genes were similar to those found in other bacteria, such as *Pseudomonas aeruginosa*, an important CF pathogen. One such gene cluster, (MAB_0295-MAB_0298), is homologous to the phenazine biosynthesis genes (*phz*) in *Streptomyces* sp. MG1 and *P. aeruginosa* PA7, and it was noted that phenazine is the precursor to pyocyanin, a *P. aeruginosa* virulence factor [20]. We show here that the *phz* gene cluster exists within a larger cluster that potentially encodes a novel secondary metabolite, and that a *phzC* deletion mutant has altered pathogenicity in the *Xenopus* model of infection. 

## 2. Materials and Methods

### 2.1. Animal Husbandry

During the experiment, all animals were handled according to the regulations of the University of Rochester Committee on Animal Resources. Tadpoles of a transgenic line, expressing a GFP reporter under control of the macrophage-specific *mpeg* promoter (*mpeg*:GFP) [21], were used for imaging. Outbred and *mpeg*:GFP tadpoles were obtained from the *X. laevis* research resource for immunology, located at the University of Rochester. All procedures involving the animals adhered strictly to the laboratory and University Committee on Animal Resources regulations (approval number 100577/2003-151).

### 2.2. Mycobacterium abscessus

Strains and Culture Conditions:

*M. abscessus* cultures were grown at 37 °C for 3 to 5 days in Middlebrook 7H9 broth with 0.2% glycerol, 10% albumin–dextrose–saline (ADS) supplement, and tyloxapol. The cultures were then centrifuged, washed with amphibian phosphate buffer (APBS), and resuspended in 50% APBS and 50% glycerol. The stocks were then frozen at −80 °C, and the number of viable colony-forming units (CFU/mL) was determined. Fluorescent DsRed Mab were generated by transformation strains using a plasmid (pMV261.Kan.DsRed), kindly provided by W. R. Jacobs (Albert Einstein College of Medicine, Bronx, NY, USA). The unmarked, in-frame Δ*phzC* deletion mutant PM3472 was constructed using a *galK* allelic exchange vector, as previously described, using PM3044 as the WT strain and a smooth morphotype of *Mab* ATCC19977, with the mutation confirmed using PCR and sequencing [22].

### 2.3. Mab Inoculation of Tadpoles and Frogs

Groups of 6 to 10 tadpoles, two (stage 50) and three weeks (stage 55; [23]) old, were (ip) inoculated with 10 μL intraperitoneal injection of 5 × 10^5^ CFU of *Mab*. One-year-old adult frogs (group of six individuals) were (ip) injected with 5 × 10^5^ CFU of *Mab* in a 100 μL volume. As a negative control, the animals were injected with the same volume of amphibian phosphate buffer (APBS). Experiments were conducted for 14 (bacterial loads and gene expression) to 50 days (survival). For tail wound assay, 2 μL of diluted DsRed WT or ∆phzC *Mab* stocks were injected into the muscle at the base of the tail, using a biopsy punch, and infection was followed for up to 7 days in individual tadpoles. 

### 2.4. Colony Formation Assay

The livers of the animals were collected and lysed by bead beating (Omni Bead Ruptor™) in 500 μL APBS. Undiluted and 10× diluted lysates (100 μL out of 500 μL total lysate) were plated on an antibiotic medium. Colony counts were obtained 5 days after sample incubation at 37 °C. Organ homogenates not used for plating were saved for DNA isolation at −80 °C.

### 2.5. Confocal Microscopy

At different times after inoculation with DsRed WT or ∆phzC *Mab* into the tail muscle of *mpeg1*:GFP tadpoles, the animals were narcotized, and the infected area was visualized under a Leica DMi8 Inverted Microscope (Wetzlar, Germany). To quantify the total volume ratio of *Mab* in the images, the Imaris microscopy image analysis software version 10.0, was used to calculate the total volume of DsRed bacteria versus the total volume of GFP-expressing macrophages.

### 2.6. Q-PCR

Animals were euthanized by tricaine anesthetic overdose. Tissues were dissected and added to TRIzol™ (Invitrogen, Waltham, MA, USA) for total RNA extraction. cDNA was synthesized using 500 ng RNA, M-MLV RT, and oligo (dT) (Invitrogen). The relative gene expression was determined using an ABI 7300 Real-Time PCR System and PerfeCTa SYBR Green FastMix (Thermo Fisher Scientific, Waltham, MA, USA), following the manufacturer’s protocol, according to the methods described in our published work [17,24,25]. For a final reaction volume of 20 μL, 2.5 μL, equivalent to 125 ng of cDNA template, was added to a mixture of 2.5 μL of each primer, 7.5 μL of the master mix buffer, and 5 μL of SYBR Green. The relative gene expression levels were determined using the ΔΔCT method [26], where expression levels were normalized to the endogenous housekeeping gene, glyceraldehyde-3-phosphate dehydrogenase (GAPDH), and then further normalized against the lowest observed expression. GAPDH was used as a housekeeping gene due to its consistent and high expression levels between the same tissues of different individuals [27], which did not significantly differ between groups in our experiments. All the primers were validated prior to use by gradient PCR, as described in the prior literature [27]. Sequences of all the primers used in the study are listed in Appendix A.

### 2.7. Statistical Analysis

For all experiments, statistical analyses were performed using GraphPad Prism 10 software (GraphPad Software, Inc., La Jolla, CA, USA), according to the experimental design of each test. Normality was tested for all experiments. For experiments with two groups, a Student’s *t*-test was used, while one-way ANOVA and Tukey’s posthoc test was used for experiments with more than two groups.

## 3. Results

Analysis of the *Mycobacterium abscessus* ATCC 19977 (*Mab*) genome revealed core mycobacterial genes and genes that we predict were obtained via horizontal gene transfer from actinobacteria [20]. One such gene cluster (MAB_0295-MAB_0298) is homologous to the phenazine biosynthesis genes (phz) in Streptomyces sp. MG1 and *P. aeruginosa* PA7 [20]. We analyzed this gene cluster in the *Mab* chromosome using Mycobrowser (https://mycobrowser.epfl.ch) and the protein annotations in UniProt (https://www.uniprot.org/uniprotkb/B1MFJ4/entry; accessed on 1 May 2024) and determined that it is part of a 23.9 kb region highly homologous to the streptonigrin biosynthesis pathway in *Streptomyces flocculus* [28]. Streptonigrin is a secondary metabolite with anti-bacterial and anti-tumor activity [29], and the pathway for its synthesis shares similarities to that of other secondary metabolites, such as calcimycin [30] and benzoxazoles [31]. Table 1 shows the predicted gene product annotations for the putative *Mab* pathway compared to the homologs within the *S. flocculus* streptonigrin pathway, while the gene cluster is shown in Figure 1. These comparisons were peformed using BLAST. The two clusters have in common an incomplete phenazine pathway lacking *phzF*, but exhibiting the ability to produce anthranilic acid, which is further modified to produce the final metabolite. In the case of the putative secondary metabolite of *Mab*, we propose that it is a simpler molecule than streptonigrin, as there are less enzymes encoded in the *Mab* cluster.

To examine the role of this putative pathway in *Mab* pathogenesis, we tested a mutant with an unmarked, in-frame deletion of the *phzC* gene in the *X. laevis* model. This work stemmed from an earlier project in which mutants were created with deletions in atypical gene clusters identified in the first *M. abscessus* genome sequencing paper, and we chose to focus on *phzC*, as it was a central gene within the originally described gene cluster.

### 3.1. Survival of Mab Infected Tadpoles

To determine the degree of pathogenicity of the *∆phzC* mutant *Mab,* we first conducted survival experiments in both tadpoles and adult frogs. We started with adult frogs that can elicit a strong inflammatory CD8 T cell-driven immune response against mycobacteria [25]. One-year-old adult frogs were inoculated by intraperitoneal injection with 5 × 10^5^ CFU of either the wild-type (WT) or *∆phzC Mab,* and their survival was monitored over a period of 50 days (Figure 2A). We observed a sharp drop in survival at 46 days in both groups. However, only a small (10%) fraction of *∆phzC*-infected *X. laevis* succumbed compared to the significantly higher fraction (70%) of frogs that died when infected with WT *Mab*. Furthermore, the *∆phzC* infected animals demonstrated a statistically reduced bacterial load compared to that of the WT *Mab* group (Figure 2B).

Since tadpoles have been shown to mount a more tolerogenic immune response to mycobacteria [25], we next infected two-week-old tadpoles (st. 50) with 5 × 10^4^ CFU of either *∆phzC* or WT S*Mab* by ip injection, and monitored survival over 50 days (Figure 2C). Similar to the results for adult frogs, there was a sharp drop in the WT infected group, with *∆phzC Mab* infected tadpoles showing a greater chance of survival over the duration of the experiment as compared with WT. The control organisms were injected with sterile amphibian phosphate buffer (APBS). 

To further investigate the pathogenicity of the *∆phzC* mutant, we determined the bacterial load in the liver of the tadpoles at 1 and 14 days post-infection. For both time points, we noticed that animals challenged with the mutant presented a lower bacterial load compared to WT, which was significantly different only at 1 dpi (Figure 3). To exclude that the 1 dpi difference between the two strains was not attributed to external factors (like pathogen concentration, inoculation method, etc.), we infected two tadpoles with the WT and *∆phzC* strains, and after 2 h, we euthanized the tadpoles and harvested their livers for colony formation assay. The bacteria were similar for both strains (Figure 3), which suggested that Phenazine deletion might affect *Mab* colonization in a tadpole liver.

### 3.2. Relative Expression Responses of Innate Immune Genes

To further investigate the pathogenicity of *∆phzC Mab*, RT-qPCR was conducted to evaluate the transcript levels of several key inflammatory cytokines (*tnfa, il1b, ifng, inos*) in the liver and lungs (Figure 4). In the lungs, significantly lower transcript levels of *IL-1β, TNF-α, iNOS,* and *IFN-γ* were found, suggesting a dampening of the tadpole immune response when infected with the ∆*phzC Mab*. In contrast, few differences were found in the liver.

### 3.3. Tracking Macrophage Recruitment and Infection Using Tail Wound Inoculation Assay

To investigate the ability of *∆phzC Mab* to induce macrophage recruitment and infection, we took advantage of the tail wound inoculation assay [19], with a transgenic line expressing a GFP reporter under control of the macrophage-specific *mpeg* promoter (*mpeg*:GFP, [21]. A total of 12 *mpeg*:GFP tadpoles were inoculated with DsRed WT or *∆phzC Mab* by biopsy punch into the base of the tail. Images were captured at three different time points (1, 3, and 7 dpi) post-infection to track macrophage recruitment to the site of infection. The site of infection for both WT and *∆phzC Mab* were well demarked with an abundant fluorescent red signal (Figure 5). However, in contrast to a marked recruitment of *mpeg*:GFP+ macrophages at the site of infection with WT *Mab* at 1 dpi, few were observed with *∆phzC Mab* infection. In addition, a large fraction of recruited macrophages was infected by DsRed WT *Mab*, as indicated by the overlay orange color. At later time points, fewer bacteria were detected at the site of infection, and fewer infected macrophages were observed. No significant increase in macrophage infiltration was detected with *∆phzC Mab* at 3 and 7 dpi (Figure 6).

To better quantify the fraction of infected macrophages, we determined the relative volume area of dual red/green fluorescence at 1, 3, and 7 dpi. The relative volume ratio of WT *Mab* was significantly higher than that of *∆phzC Mab* at 1 dpi and tended to decrease at later time points (Figure 7).

## 4. Discussion

The goal of this study was to assess the impact of deleting the putative virulence gene within a newly recognized putative secondary metabolite pathway on *Mab* pathogenicity in *X. laevis.* Our data suggest that the *∆phzC Mab* is indeed less virulent than the WT *Mab* control, inducing lower mortality in both adult frogs and tadpoles. In addition, *∆phzC Mab* infection is characterized by lower bacterial loads at early stages (1 dpi), leading to reduced inflammatory gene expression response and poor macrophage recruitment at the site of infection. The lower bacterial loads and reduced macrophage recruitment suggest some defect of *∆phzC Mab* mutants in colonizing the host and possibly interacting with macrophages. 

It is interesting to note that infection with *∆phzC Mab* mutants is less lethal for both adult frogs, able to mount inflammatory T cell response, and tadpoles, eliciting a more tolerogenic immune response. In a previous study using *Mycobacterium marinum* infection, we showed that adult frogs exhibited a strong inflammatory immune response driven by CD8 T cells, whereas the tadpole immune response was minimally inflammatory, mainly relying on innate-like T cells [19]. It will be interesting to further compare the immune response of WT and *∆phzC Mab* between adults and tadpoles.

To examine in vivo immune cell recruitment, as well as cells infected by *Mab* and the dissemination of infected cells in a localized area suitable for intravital microscopy, we used a tail inoculation assay that was first adapted for studying the inflammatory response following injury [21] and then optimized for *M. marinum* infection [19]. The system takes advantage of a *X. laevis* transgenic strain expressing fluorescent reporters in immune cell populations, including cells of the myeloid lineages (GFP controlled by the *lurp* promoter) and stably transfected WT and *∆phzC Mab* bacteria, expressing DsRed as a fluorescent reporter. By using real-time confocal microscopy, we detected numerous mpeg^+^ macrophages at the site within 1 dpi, and a large fraction of these macrophage were infected with WT DsRed *Mab*. This contrasted with the markedly reduced number of mpeg^+^ macrophages recruited and infected by *∆phzC Mab*. This difference was unlikely due to a slower kinetics, since the number of mpeg^+^ macrophages remained scarce at 3 and 7 dpi. Furthermore, the detection of mpeg^+^ macrophages infected with DsRed *∆phzC Mab* suggests that the mutation does not significantly impact the ability of Mab to infect macrophages, but rather diminishes the recruitment of these cells, which is consistent with the drop in *∆phzC Mab* at the early stage of ip infection.

The *phzC* gene is located in a gene cluster, mostly syntenic with that of the Streptomyces streptonigrin pathway; however, the streptonigrin pathway is larger, encompassing 48 genes, while that of the cluster in Mab consists of 17 genes, of which 12 are shared between the two clusters. The five unique genes in *Mab* include four putative enzymes and one transcriptional regulator. We propose that the *Mab* gene cluster produces a molecule less complex than that of streptonigrin. Given that streptonigrin has been shown to interfere with b-catenin/Tcf signaling, we speculate that perhaps a structurally similar compound produced by *Mab* could affect this pathway [29].

In conclusion, our data provide evidence that *phzC* deficiency reduces *Mab* pathogenicity and macrophage recruitment in *X. laevis*. As such, the *phzC* gene and the related secondary metabolite pathway may be of relevance for *Mab* pathogenesis in humans and may serve as a new therapeutic target.

## Figures and Tables

**Figure 1 microorganisms-12-01120-f001:**
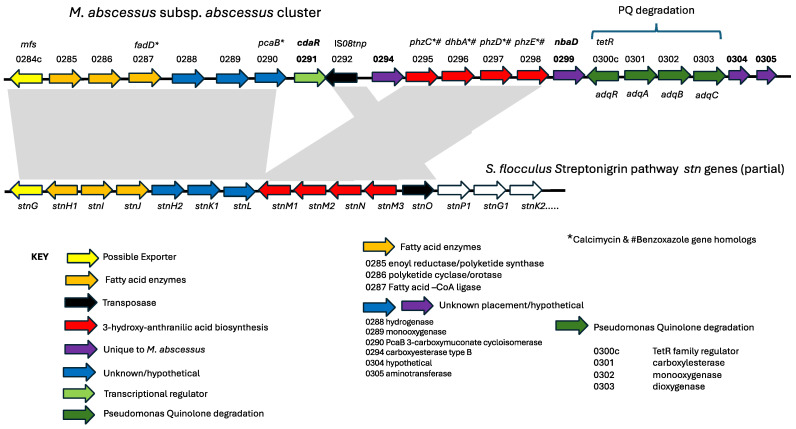
Putative secondary metabolite pathway in *M. abscessus*. The cluster of genes (MAB_0284c to MAB_0305) is shown at the top, with the MAB ORF number designations from the genome of *M. abscessus* ATTC 19977. Embedded within the cluster are MAB_0330c to MAB_0303, also known as the *adqRABC* cluster, which has been shown to be responsible for the degradation of quinolone signaling molecules from *P. aeruginosa* [31]. The MAB_0284c to MAB_0290 genes are syntenic with those in the streptonigrin pathway genes (*stnG* to *stnL*), as shown below the *Mab* cluster. MAB_0290 and MAB_0295 to MAB_0298 are also syntenic with the streptonigrin genes, except that they are inverted in *Mab* and include two additional genes, a transcriptional regulator (MAB_0291, *cdaR*) and a putative carboxyesterase (MAB_0294). *Mab* gene products that are homologous to those in the calcimycin and benzoxazole pathways are indicated (* or #). Genes unique to the Mab cluster are shown in bold: MAB_0291 (*cdaR*), MAB_0294 (putative carboxyesterase), MAB_0299 (*nbaD*, a putative aminohydrolase), MAB_0304 (hypothetical), and MAB_0305 (putative aminotransferase). Predicted functions of some of the gene products are color-coded in the key.

**Figure 2 microorganisms-12-01120-f002:**
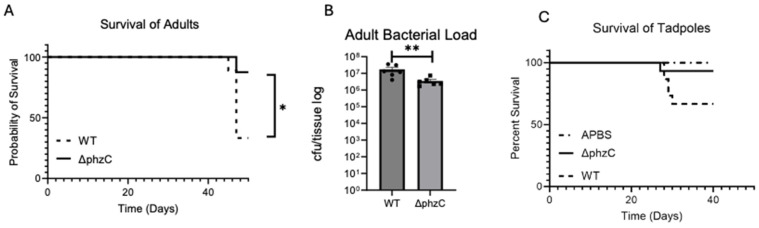
Survival of adult frogs and tadpoles following ip inoculation with either WT *Mab* or *∆phzC Mab.* (**A**) One-year-old frogs (six per group) were (ip) inoculated with 5 × 10^5^ CFU of WT or *∆phzC Mab* and monitored for 50 days for survival. (**B**) The bacterial loads in livers of adult frogs from **A**, determined at the end of the experiment. (**C**) Two-week-old tadpoles (st. 44–45, 10 per group), were inoculated with 5 × 10^5^ CFU of WT or *∆phzC Mab*. As a negative control, tadpoles were injected with APBS. Survival was monitored for 50 days for survival. Statistical differences between groups were assessed by a log-rank (Mantel–Cox) test (* *p* > 0.05; ** *p* > 0.001).

**Figure 3 microorganisms-12-01120-f003:**
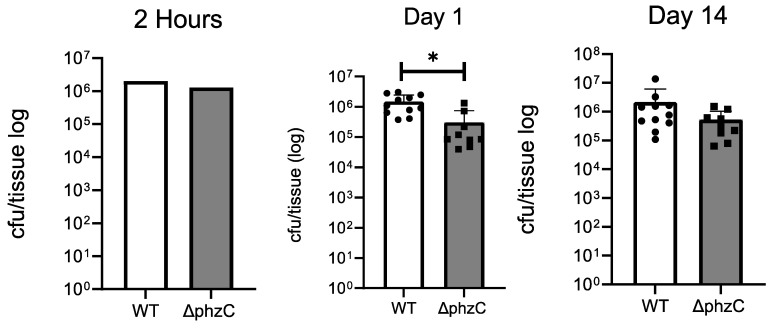
Changes in bacterial loads in tadpoles following (ip) inoculation with WT or *∆phzC Mab*. Three-weeks-old tadpoles (stage 55; 10 per group) were (ip) inoculated with 5 × 10^5^ CFU of either WT or *∆phzC Mab*. Colony assays was performed on liver lysates at 1 and 14 days post-infection. One tadpole per condition was also tested 2 h post-infection to control the initial infection load. Statistical differences between the groups was determined by *t*-test (* *p* > 0.05).

**Figure 4 microorganisms-12-01120-f004:**
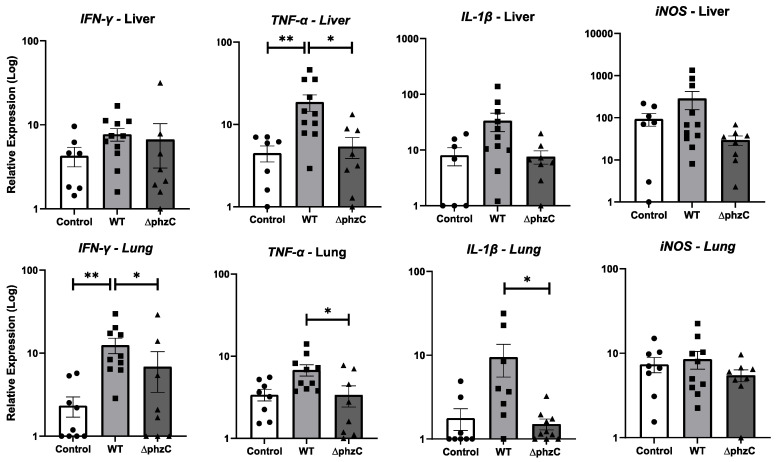
Change in relative gene expression of tadpole livers and lungs at 14 days post-infection with WT or *∆phzC Mab.* Three-week-old tadpoles (stage 55; 10 per group) were (ip) inoculated with 5 × 10^5^ CFU of either WT or *∆phzC Mab,* or injected with APBS as a negative control (Control). RNAs from the collected livers and lungs were used to determine gene expression for TNF-α, IL-1β, IFN-I, iNOS, and IFN-γ. Gene relative quantification (RQ) was determined as the fold increase relative to the GAPDH endogenous control, normalized to the lowest expression level. Bars represent standard deviations (n = 10 animals). Statistical differences between the groups were determined by one-way ANOVA and Tukey’s post hoc test. (**) *p* > 0.001; (*) *p* > 0.05.

**Figure 5 microorganisms-12-01120-f005:**
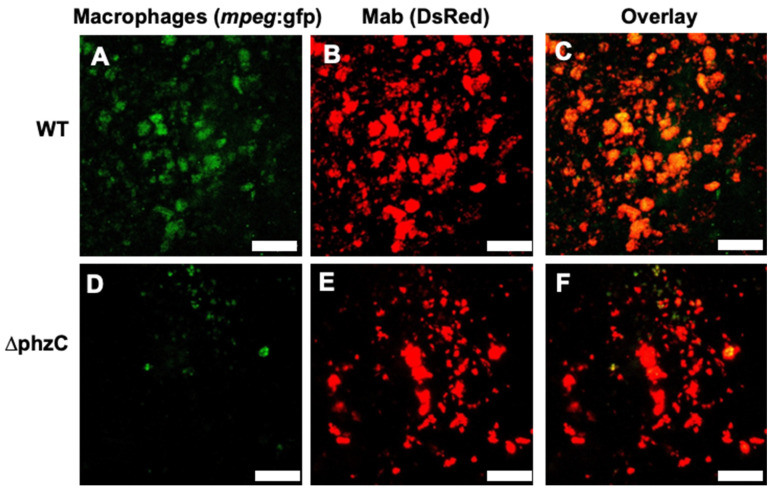
Visualization by intravital confocal microscopy of macrophage recruitment and infection in representative tadpoles 1 day after tail wound inoculation. DsRed WT (**A**–**C**) or *∆phzC Mab* (**D**–**F**) were inoculated using a biopsy punch in the tail muscle of *mpeg:GFP* transgenic tadpoles (stage 56). Infiltration of mpeg1^+^ macrophage (green) and mycobacteria-infected macrophages (red^+^ green^+^) were monitored in live tadpoles at 1 dpi using confocal microscopy (Leica DMi8 Inverted Microscope). GFP only (**A**,**D**), DsRed only (**B**,**E**), and overlayed (**C**,**F**) signals are shown. Size bar: 10 μm.

**Figure 6 microorganisms-12-01120-f006:**
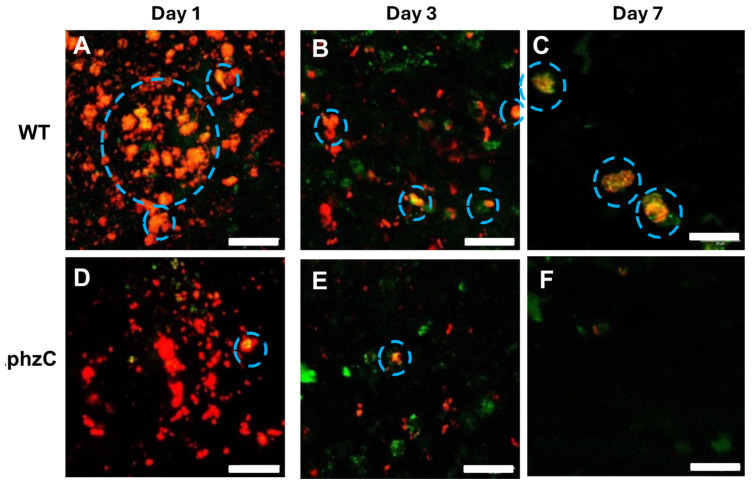
Visualization of macrophage recruitment and infection following 1, 3, and 7 days following tadpole tail wound inoculation using intravital confocal microscopy. DsRed WT (**A**–**C**) or *∆phzC* (**D**–**F**) *Mab* were inoculated with a biopsy punch in the tail muscle of *mpeg:GFP* transgenic tadpoles (stage 56). GFP mpeg1^+^ macrophage (green) and DsRed mycobacteria-infected macrophages (red^+^ green^+^) were monitored in live tadpoles at 1, 3, and 7 dpi by confocal microscopy (Leica DMi8 Inverted Microscope). Overlayed (red^+^/green^+^) signal is shown. Infected macrophages are circled. Size bar: 10 μm.

**Figure 7 microorganisms-12-01120-f007:**
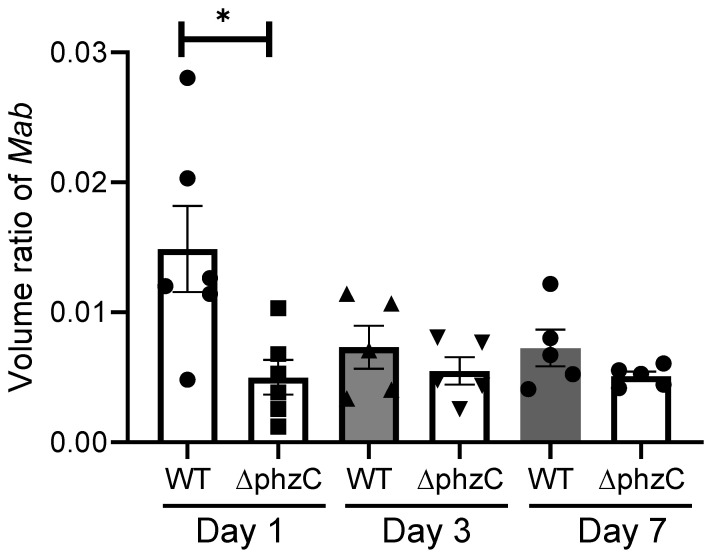
Volume ratio of DsRed Mab determined by confocal microscopy using the Imaris microscopy image analysis software to calculate the total volume of DsRed bacteria versus the total volume of GFP-expressing macrophages. The volume ratio was calculated for each tadpole tail wound from sections that totaled between ~1–2 square millimeters. N = 5 per group. (*) *p* > 0.05.

**Table 1 microorganisms-12-01120-t001:** Comparison between the MAB_0284c to MAB_0305 gene cluster and the Streptonigrin *stn* biosynthesis genes.

MAB_ORF#	*stn* Gene	Annotation/Description	E-Value *	% Identity/Similarity *
0284c	*stnG*	Major facilitator transporter	1 × 10^−19^	29/50
0285	*stnH1*	Oxidoreductase, enoylreductase	4 × 10^−48^	51/63
0286	*stnI*	Polyketide cyclase/dehydrase	4 × 10^−54^	54/69
0287	*stnJ*	FadD, fatty acid—CoA ligase	0.0	64/76
0288	*stnH2*	Monooxygenase	5 × 10^−127^	53/63
0289	*stnK1*	Hydroxylase/monooxygenase	0.0	64/74
0290	*stnL*	PcaB, 3-carboxymuconate cycloisomerase	2 × 10^−155^	58/69
0291	NP	CdaR, transcriptional regulator	NA	NA
0292	*stnO*	ISXo8 transposase	2 × 10^−61^	35/42
0294	NP	Carboxyesterase type B	NA	NA
0295	*stnM3*	PhzC, 2-dehydro-3-deoxyheptonate synthase	7 × 10^−158^	59/69
0296	*stnM2*	PhzD, isochorismatase	6 × 10^−66^	56/65
0297	*stnN*	DhabA, 2,3-dihydroxybenzoate-2,3-dehydrogenase	2 × 10^−92^	63/74
0298	*stnM1*	PhzE, anthranilate synthase	0.0	64/74
0299	NP	NbaD, aminocarboxymuconate-semialdehyde decarboxylase	NA	NA
0304	NP	Hypothetical	NA	NA
0305	NP	Aminotransferase	NA	NA

NP = not present in streptonigrin cluster; NA = not applicable; * results from BLASTP analysis [32].

## Data Availability

The raw data supporting the conclusions of this article will be made available by the authors on request.

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
