# Peer review of "Exploring the Role of a Putative Secondary Metabolite Biosynthesis Pathway in Mycobacterium abscessus Pathogenesis Using a Xenopus laevis Tadpole Model"

_microorganisms, 2024, doi:10.3390/microorganisms12061120_

Round 1

Reviewer 1 Report

Comments and Suggestions for Authors

Researchers have identified a potential secondary metabolite pathway within Mycobacterium abscessus (Mab), which could be crucial for its pathogenesis. An in-frame deletion of the MAB_0295 (phzC) gene was created and tested in a Xenopus laevis model, which has been validated for studying persistent Mab infection and pathogenesis. The study revealed that tadpoles infected with the ΔphzC mutant had lower initial bacterial loads and significantly improved survival rates compared to those infected with wild-type Mab. The ΔphzC mutant also induced lower levels of several pro-inflammatory cytokines in the liver and lungs compared to the wild type. Furthermore, macrophage recruitment and infection were reduced in tadpoles inoculated with the ΔphzC mutant as shown by intravital confocal microscopy. These findings underscore the utility of X. laevis tadpoles as a novel model for identifying genetic determinants of Mab immune-pathogenesis and indicate a role for this new and uncharacterized pathway in Mab pathogenesis and macrophage recruitment.

Major points:

In the Materials and Methods section, the chapter “2.3. Mab inoculation of tadpoles and frogs” should be supplemented with important details such as the number of adult frogs and tadpoles inoculated, as well as the duration of the experiment.

The author states in the caption of the second figure that survival was monitored for 60 days, but there are no data related to the 60th day on the figure. Please complete this, explain the reason for the omission, or modify the text accordingly.

In line 211, the author mentions IFN-I. The reviewer is confused because IFN-g was mentioned earlier in the manuscript. IFN-g is a type II interferon. The IFN-I group mentioned by the author includes, for example, IFN-a, and IFN-b. Please clarify which one was studied.

Author writes in the line 255 “The relative volume ratio of WT Mab was significantly higher than ∆phzC Mab at 1 dpi and significantly decreased at later time points” What does the second part of this sentence mean? The figure does not indicate further significance related to this.

During their research, the author conducted experiments with both adult frogs and tadpoles and presented the beneficial effects of the Mab DphzC mutation in frogs of both age groups. However, they did not provide data on inflammatory cytokine expression in lungs and liver for the adult frogs, only for the tadpoles. The reviewer believes that presenting this data could better support the use of X. laevis in studying Mab pathogenesis. I recommend presenting these data or deleting the data specific for adult frogs in the manuscript.

Minor points:

In line 63: instead of Xenopus laevis use X. laevis

In line 85 and line: Use a consistent notation for indicating degrees Celsius. In the notation for degrees Celsius, in line 85 the small circle is in the superscript, whereas in line 89 it is in the middle. Check the line 107, 108.

In the description of Materials and Methods, use a consistent style of writing. Either capitalize each relevant word or only the first word of the title. Currently, it is used inconsistently. For example: Mab inoculation of tadpoles and frogs, Statistical analysis”/ “Animal Husbandry, Colony Formation Assay”

In line 98, 99 105: Use a consistent symbol for the multiplication sign throughout the manuscript. Sometimes a large 'X' is used, and other times a small 'x' is found.

In line 128-130: The sentence found in lines 128-130 is identical to a sentence in the introduction (line 67-69). Please avoid repetitions and rephrase it.

Use the term "Mycobacterium abscessus subs abscessus" in the title of the first figure caption. This term is not used elsewhere, and should be explained in the manuscript.

Please revise the second figure both in content and format. In panel "A", include the data for the adults injected with APBS as a negative control. For panel "B", use the formatting of panels "A" and "C", meaning capitalize the first letter of each word in the title and on the y-axis labels. The "B" panel uses labels on the x-axis that have not been used before and are not explained anywhere. Replace "Phz-" with "DPhzC", and the "S" designation is unclear to the reviewer; perhaps it refers to "WT"? For the amphibian phosphate buffer notation, use the designation used in the manuscript in panel "C" (APBS instead of aBPS).

Please rephrase the title of Figure 2 so that it reflects the order of the panels (use Survival of the adult frogs and tadpoles instead of “Survival of tadpoles and adult frogs…”)

In line 189: use APBS instead of APBs

In line 209: Please delete the word "ifng" as it is duplicated.

In the panels of Figure 4, use a consistent method for writing the words "lung" and "liver" — either in italics or not, but do not mix styles.

In the legend of Figure 4, delete the letter "S" in line 216, and the abbreviation "(Un)" after "negative control" in line 217, as these do not appear on the figure and are not explained.

In line 218, please delete the term "IFN-I," and in the following line, correct it to "IFN-g," as I believe that is correct.

In the Results section, please write the titles of the subsections consistently with either all words starting with a capital letter or all in lowercase, to ensure uniformity. (Here, you can find 3 examples from the manuscript: Survival of Mab Infected Tadpoles versus Relative expression responses of innate immune genes versus Tracking Macrophage Recruitment and infection using tail wound inoculation assay)

In line 228, 230, 233, 237 please italicize the abbreviation "DphzC"

The title of the legend for Figure 5 is missing the information about how many days after the infection the visualizations were taken.

In the captions for Figures 5 and 6, standardize the use of the "+" sign, either as a superscript or not, but ensure it is consistent.

In line 264, 269 please italicize the abbreviation "DphzC"

Comments on the Quality of English Language

Reviewer 2 Report

Comments and Suggestions for Authors

This is an important topic with clinical significance;  moreover the authors employ a model system that has decided benefits compared to the murine and zebrafish model systems.  The paper reports an interesting result but it is int he category of a "brief reports" paper rather than a thorough analysis of the finding.  Even so, there were many aspects of the findings that were difficult to interpret given the very brief descriptions provided. There are several concerns:  1.  The results are quite limited - for mutation of a single gene with no analysis of the pathway;  fine for s "brief report" but not a full study;  2.  some of the methods are questionable - e.e. SYBR green for real time PCR; 3. The methods are not  explained in sufficient detail to analyze the findings and assess the results.  Even as a brief report the paper could be strengthened by addressing the following comments/questions. 

l46:  the authors mention the rough phenotype; this should be described in more detail; did the authors observe this in their study?  this also could affect the phenotypes they observed. What percentage of the bacterial cells adopted this phenotype?

l 80: the authors describe the mpeg:GFP tadpoles in more detail 

l 131:  the authors should describe the bioinformatics they used to analyze the genome and discover the phenazine biosynthesis 131 genes (phz) cluster

l  165  need more detail on why they mutated the gene phzC gene and only this one in the larger pathway; the paper would have been stronger if they analyzed more genes in the pathway

The authors should describe their choice of statistical tests and the rationale in more detail in the material and methods section

Fig 2;  the authors should show individual samples (n)

Moreover, the survival graphs are not very clear;  they need a different formatfor displaying total amount of data for these

Fig 2 graph B is not very informative - need greater spread near the point in the range where the samples are counted

What is adult 1st replica?  this is note explained well at all in the text

Fig 3:  why only 1 tadpole per condition?  this is not very revealing or representative.

Fig 5:  These figures would benefit by a brightfield image as well to show some sort of tissue/cellular resolution.  Just looking a the fluor. signal is not that informative.

How was relative volume calculated? confocal measurements using all three dimensions?  What area of tissue? Much more detail is required to assess these numbers. 

SYBR green is notoriously variable and imprecise; why not use a taqman assay?  This would give way more confidence in the results.  

The authors reported using realtime confocal microscopy -- were these analyzed in real time?   Is each tadpole followed over time individually?   Or are the data aggregated?  Way more detail on the confocal microscopy is necessary.

Minor edits: l40:  avoid " a lot" - too colloquial

l70  avoid "like"   similar to perhaps ...

Round 2

Reviewer 1 Report

Comments and Suggestions for Authors

I accept the manuscript in its present form.

Comments on the Quality of English Language

I accept the manuscript in its present form.

Reviewer 2 Report

Comments and Suggestions for Authors

The reviewers were responsive to the points that did not require any additional wet lab work and this has definitely improved the manuscript.  This would be an ideal paper for a "brief reports"  given the focus on only a few genes.